# Deep Generative Models with Learnable Knowledge Constraints

**Zhiting Hu,  Zichao Yang,  Ruslan Salakhutdinov,**
**Xiaodan Liang,  Lianhui Qin,  Haoye Dong,  Eric P. Xing**
Carnegie Mellon University,   Petuum Inc.
{zhitingh,zichaoy,rsalakhu,xiaodan1}@cs.cmu.edu, eric.xing@petuum.com

## Abstract

The broad set of deep generative models (DGMs) has achieved remarkable advances. However, it is often difficult to incorporate rich structured domain knowledge with the end-to-end DGMs. Posterior regularization (PR) offers a principled framework to impose structured constraints on probabilistic models, but has limited applicability to the diverse DGMs that can lack a Bayesian formulation or even explicit density evaluation. PR also requires constraints to be fully specified *a priori*, which is impractical or suboptimal for complex knowledge with learnable uncertain parts. In this paper, we establish mathematical correspondence between PR and reinforcement learning (RL), and, based on the connection, expand PR to learn constraints as the extrinsic reward in RL. The resulting algorithm is model-agnostic to apply to any DGMs, and is flexible to adapt arbitrary constraints with the model jointly. Experiments on human image generation and templated sentence generation show models with learned knowledge constraints by our algorithm greatly improve over base generative models.

## 1   Introduction

Generative models provide a powerful mechanism for learning data distributions and simulating samples. Recent years have seen remarkable advances especially on the deep approaches [16, 25] such as Generative Adversarial Networks (GANs) [15], Variational Autoencoders (VAEs) [27], auto-regressive networks [29, 42], and so forth. However, it is usually difficult to exploit in these various deep generative models rich problem structures and domain knowledge (e.g., the human body structure in image generation, Figure 1). Many times we have to hope the deep networks can discover the structures from massive data by themselves, leaving much valuable domain knowledge unused. Recent efforts of designing specialized network architectures or learning disentangled representations [5, 23] are usually only applicable to specific knowledge, models, or tasks. It is therefore highly desirable to have a *general* means of incorporating arbitrary structured knowledge with any types of deep generative models in a principled way.

On the other hand, posterior regularization (PR) [13] is a principled framework to impose knowledge constraints on posterior distributions of probabilistic models, and has shown effectiveness in regulating the learning of models in different context. For example, [21] extends PR to incorporate structured logic rules with neural classifiers. However, the previous approaches are not directly applicable to the general case of deep generative models, as many of the models (e.g., GANs, many auto-regressive networks) are not straightforwardly formulated with the probabilistic Bayesian framework and do not possess a posterior distribution or even meaningful latent variables. Moreover, PR has required *a priori* fixed constraints. That means users have to fully specify the constraints beforehand, which can be impractical due to heavy engineering, or suboptimal without adaptivity to the data and models. To extend the scope of applicable knowledge and reduce engineering burden, it is necessary to allow

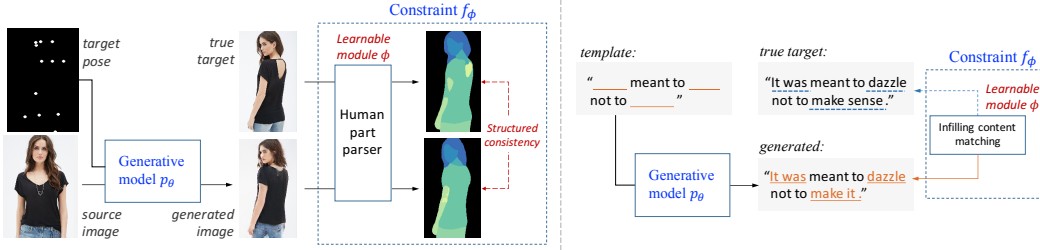

**Figure 1:** Two example applications of imposing learnable knowledge constraints on generative models. **Left:** Given a person image and a target pose (defined by key points), the goal is to generate an image of the person under the new pose. The constraint is to force the human parts (e.g., head) of the generated image to match those of the true target image. **Right:** Given a text template, the goal is to generate a complete sentence following the template. The constraint is to force the match between the infilling content of the generated sentence with the true content. (See sec 5 for more details.)

users to specify only *partial* or *fuzzy* structures, while learning remaining parts of the constraints jointly with the regulated model.

To this end, we establish formal connections between the PR framework with a broad set of algorithms in the control and reinforcement learning (RL) domains, and, based on the connections, transfer well-developed RL techniques for constraint learning in PR. In particular, though the PR framework and the RL are apparently distinct paradigms applied in different context, we show mathematical correspondence between the model and constraints in PR with the policy and reward in entropy-regularized policy optimization [43, 45, 1], respectively. This thus naturally inspires to leverage relevant approach from the RL domain (specifically, the maximum entropy inverse RL [56, 11]) to learn the PR constraints from data (i.e., demonstrations in RL).

Based on the unified perspective, we drive a practical algorithm with efficient estimations and moderate approximations. The algorithm is efficient to regularize large target space with arbitrary constraints, flexible to couple adapting the constraints with learning the model, and model-agnostic to apply to diverse deep generative models, including implicit models where generative density cannot be evaluated [40, 15]. We demonstrate the effectiveness of the proposed approach in both image and text generation (Figure 1). Leveraging domain knowledge of structure-preserving constraints, the resulting models improve over base generative models.

## 2 Related Work

It is of increasing interest to incorporate problem structures and domain knowledge in machine learning approaches [49, 13, 21]. The added structure helps to facilitate learning, enhance generalization, and improve interpretability. For deep neural models, one of the common ways is to design specialized network architectures or features for specific tasks (e.g., [2, 34, 28, 33]). Such a method typically has a limited scope of applicable tasks, models, or knowledge. On the other hand, for structured probabilistic models, posterior regularization (PR) and related frameworks [13, 32, 4] provide a general means to impose knowledge constraints during model estimation. [21] develops *iterative knowledge distillation* based on PR to regularize neural networks with any logic rules. However, the application of PR to the broad class of deep generative models has been hindered, as many of the models do not even possess meaningful latent variables or explicit density evaluation (i.e., implicit models). Previous attempts thus are limited to applying simple max-margin constraints [31]. The requirement of *a priori* fixed constraints has also made PR impractical for complex, uncertain knowledge. Previous efforts to alleviate the issue either require additional manual supervision [39] or is limited to regularizing small label space [22]. This paper develops a *practical* algorithm that is generally applicable to any deep generative models and any learnable constraints on arbitrary (large) target space.

Our work builds connections between the Bayesian PR framework and reinforcement learning. A relevant, broad research topic of formalizing RL as a probabilistic inference problem has been explored in the RL literature [6, 7, 41, 30, 1, 48], where rich approximate inference tools are used to improve the modeling and reasoning for various RL algorithms. The link between RL and PR

| Components | PR | Entropy-Reg RL | MaxEnt IRL | (Energy) GANs |
|---|---|---|---|---|
| $\boldsymbol{x}$ | data/generations | action-state samples | demonstrations | data/generations |
| $p(\boldsymbol{x})$ | generative model $p_\theta$ | (old) policy $p_\pi$ | — | generator |
| $f(\boldsymbol{x})/R(\boldsymbol{x})$ | constraint $f_\phi$ | reward $R$ | reward $R_\phi$ | discriminator |
| $q(\boldsymbol{x})$ | variational distr. $q$, Eq.3 | (new) policy $q_\pi$ | policy $q_\phi$ | — |

**Table 1:** Unified perspective of the different approaches, showing mathematical correspondence of PR with the entropy-regularized RL (sec 3.2.1) and maximum entropy IRL (sec 3.2.2), and its (conceptual) relations to (energy-based) GANs (sec 4).

has not been previously studied. We establish the mathematical correspondence, and, differing from the RL literature, we in turn transfer the tools from RL to expand the probabilistic PR framework. Inverse reinforcement learning (IRL) seeks to learn a reward function from expert demonstrations. Recent approaches based on maximum-entropy IRL [56] are developed to learn both the reward and policy [11, 10, 12]. We adopt the maximum-entropy IRL formulation to derive the constraint learning objective in our algorithm, and leverage the unique structure of PR for efficient importance sampling estimation, which differs from these previous approaches.

# 3 Connecting Posterior Regularization to Reinforcement Learning

## 3.1 PR for Deep Generative Models

PR [13] was originally proposed to provide a principled framework for incorporating constraints on posterior distributions of probabilistic models with latent variables. The formulation is not generally applicable to deep generative models as many of them (e.g., GANs and autoregressive models) are not formulated within the Bayesian framework and do not possess a valid posterior distribution or even semantically meaningful latent variables. Here we adopt a slightly adapted formulation that makes minimal assumptions on the specifications of the model to regularize. It is worth noting that though we present in the generative model context, the formulations, including the algorithm developed later (sec 4), can straightforwardly be extended to other settings such as discriminative models.

Consider a generative model $\boldsymbol{x} \sim p_\theta(\boldsymbol{x})$ with parameters $\boldsymbol{\theta}$. Note that generation of $\boldsymbol{x}$ can condition on arbitrary other elements (e.g., the source image for image transformation) which are omitted for simplicity of notations. Denote the original objective of $p_\theta(\boldsymbol{x})$ with $\mathcal{L}(\boldsymbol{\theta})$. PR augments the objective by adding a constraint term encoding the domain knowledge. Without loss of generality, consider constraint function $f(\boldsymbol{x}) \in \mathbb{R}$, such that a higher $f(\boldsymbol{x})$ value indicates a better $\boldsymbol{x}$ in terms of the particular knowledge. Note that $f$ can also involve other factors such as latent variables and extra supervisions, and can include a set of multiple constraints.

A straightforward way to impose the constraint on the model is to maximize $\mathbb{E}_{p_\theta}[f(\boldsymbol{x})]$. Such method is efficient only when $p_\theta$ is a GAN-like implicit generative model or an explicit distribution that can be efficiently reparameterized (e.g., Gaussian [27]). For other models such as the large set of non-reparameterizable explicit distributions, the gradient $\nabla_\theta \mathbb{E}_{p_\theta}[f(\boldsymbol{x})]$ is usually computed with the *log-derivative* trick and can suffer from high variance. For broad applicability and efficient optimization, PR instead imposes the constraint on an auxiliary variational distribution $q$, which is encouraged to stay close to $p_\theta$ through a KL divergence term:

$$\mathcal{L}(\boldsymbol{\theta}, q) = \mathrm{KL}(q(\boldsymbol{x}) \| p_\theta(\boldsymbol{x})) - \alpha \mathbb{E}_q[f(\boldsymbol{x})], \tag{1}$$

where $\alpha$ is the weight of the constraint term. The PR objective for learning the model is written as:

$$\min_{\theta, q} \mathcal{L}(\boldsymbol{\theta}) + \lambda \mathcal{L}(\boldsymbol{\theta}, q), \tag{2}$$

where $\lambda$ is the balancing hyperparameter. As optimizing the original model objective $\mathcal{L}(\boldsymbol{\theta})$ is straightforward and depends on the specific generative model of choice, in the following we omit the discussion of $\mathcal{L}(\boldsymbol{\theta})$ and focus on $\mathcal{L}(\boldsymbol{\theta}, q)$ introduced by the framework.

The problem is solved using an EM-style algorithm [13, 21]. Specifically, the E-step optimizes Eq.(1) w.r.t $q$, which is convex and has a closed-form solution at each iteration given $\boldsymbol{\theta}$:

$$q^*(\boldsymbol{x}) = p_\theta(\boldsymbol{x}) \exp\{\alpha f(\boldsymbol{x})\} / Z, \tag{3}$$

where $Z$ is the normalization term. We can see $q^*$ as an energy-based distribution with the negative energy defined by $\alpha f(\boldsymbol{x}) + \log p_\theta(\boldsymbol{x})$. With $q$ from the E-step fixed, the M-step optimizes Eq.(1) w.r.t $\boldsymbol{\theta}$ with:

$$\min_\theta \text{KL}(q(\boldsymbol{x})\|p_\theta(\boldsymbol{x})) = \min_\theta -\mathbb{E}_q\left[\log p_\theta(\boldsymbol{x})\right] + const. \tag{4}$$

Constraint $f$ in PR has to be fully-specified *a priori* and is fixed throughout the learning. It would be desirable or even necessary to enable learnable constraints so that practitioners are allowed to specify only the known components of $f$ while leaving any unknown or uncertain components automatically learned. For example, for human image generation in Figure 1, left panel, users are able to specify structures on the parsed human parts, while it is impractical to also manually engineer the human part parser that involves recognizing parts from raw image pixels. It is favorable to instead cast the parser as a learnable module in the constraint. Though it is possible to pre-train the module and simply fix in PR, the lack of adaptivity to the data and model can lead to suboptimal results, as shown in the empirical study (Table 2). This necessitates to expand the PR framework to enable joint learning of constraints with the model.

Denote the constraint function with learnable components as $f_\phi(\boldsymbol{x})$, where $\phi$ can be of various forms that are optimizable, such as the free parameters of a structural model, or a graph structure to optimize.

**Simple way of learning the constraint.** A straightforward way to learn the constraint is to directly optimize Eq.(1) w.r.t $\phi$ in the M-step, yielding

$$\max_\phi \mathbb{E}_{\boldsymbol{x}\sim q(\boldsymbol{x})}[f_\phi(\boldsymbol{x})]. \tag{5}$$

That is, the constraint is trained to fit to the samples from the current regularized model $q$. However, such objective can be problematic as the generated samples can be of low quality, e.g., due to poor state of the generative parameter $\boldsymbol{\theta}$ at initial stages, or insufficient capability of the generative model per se.

In this paper, we propose to treat the learning of constraint as an *extrinsic reward*, as motivated by the connections between PR with the reinforcement learning domain presented below.

## 3.2 PR and RL

RL or optimal control has been studied primarily for determining optimal action sequences or strategies, which is significantly different from the context of PR that aims at regulating generative models. However, formulations very similar to PR (e.g., Eqs.1 and 3) have been developed and widely used, in both the (forward) RL for policy optimization and the inverse RL for reward learning.

To make the mathematical correspondence clearer, we intentionally re-use most of the notations from PR. Table 1 lists the correspondence. Specifically, consider a stationary Markov decision process (MDP). An agent in state $s$ draws an action $a$ following the policy $p_\pi(a|s)$. The state subsequently transfers to $s'$ (with some transition probability of the MDP), and a reward is obtained $R(s, a) \in \mathbb{R}$. Let $\boldsymbol{x} = (s, a)$ denote the state-action pair, and $p_\pi(\boldsymbol{x}) = \mu^\pi(s)p_\pi(a|s)$ where $\mu^\pi(s)$ is the stationary state distribution [47].

### 3.2.1 Entropy regularized policy optimization

The goal of policy optimization is to find the optimal policy that maximizes the expected reward. The rich research line of entropy regularized policy optimization has augmented the objective with information theoretic regularizers such as KL divergence between the new policy and the old policy for stabilized learning. With a slight abuse of notations, let $q_\pi(\boldsymbol{x})$ denote the new policy and $p_\pi(\boldsymbol{x})$ the old one. A prominent algorithm for example is the relative entropy policy search (REPS) [43] which follows the objective:

$$\min_{q_\pi} \mathcal{L}(q_\pi) = \text{KL}(q_\pi(\boldsymbol{x})\|p_\pi(\boldsymbol{x})) - \alpha\mathbb{E}_{q_\pi}\left[R(\boldsymbol{x})\right], \tag{6}$$

where the KL divergence prevents the policy from changing too rapidly. Similar objectives have also been widely used in other workhorse algorithms such as trust-region policy optimization (TRPO) [45], soft Q-learning [17, 46], and others.

We can see the close resemblance between Eq.(6) with the PR objective in Eq.(1), where the generative model $p_\theta(\boldsymbol{x})$ in PR corresponds to the reference policy $p_\pi(\boldsymbol{x})$, while the constraint $f(\boldsymbol{x})$ corresponds

to the reward $R(\boldsymbol{x})$. The new policy $q_\pi$ can be either a parametric distribution [45] or a non-parametric distribution [43, 1]. For the latter, the optimization of Eq.(6) precisely corresponds to the E-step of PR, yielding the optimal policy $q_\pi^*(\boldsymbol{x})$ that takes the same form of $q^*(\boldsymbol{x})$ in Eq.(3), with $p_\theta$ and $f$ replaced with the respective counterparts $p_\pi$ and $R$, respectively. The parametric policy $p_\pi$ is subsequently updated with samples from $q_\pi^*$, which is exactly equivalent to the M-step in PR (Eq.4).

While the above policy optimization algorithms have assumed a reward function given by the external environment, just as the pre-defined constraint function in PR, the strong connections above inspire us to treat the PR constraint as an extrinsic reward, and utilize the rich tools in RL (especially the inverse RL) for learning the constraint.

### 3.2.2 Maximum entropy inverse reinforcement learning

Maximum entropy (MaxEnt) IRL [56] is among the most widely-used methods that induce the reward function from expert demonstrations $\boldsymbol{x} \sim p_d(\boldsymbol{x})$, where $p_d$ is the empirical demonstration (data) distribution. MaxEnt IRL adopts the same principle as the above entropy regularized RL (Eq.6) that maximizes the expected reward regularized by the relative entropy (i.e., the KL), except that, in MaxEnt IRL, $p_\pi$ is replaced with a *uniform* distribution and the regularization reduces to the entropy of $q_\pi$. Therefore, same as above, the optimal policy takes the form $\exp\{\alpha R(\boldsymbol{x})\}/Z$. MaxEnt IRL assumes the demonstrations are drawn from the optimal policy. Learning the reward function $R_\phi(\boldsymbol{x})$ with unknown parameters $\phi$ is then cast as maximizing the likelihood of the distribution $q_\phi(\boldsymbol{x}) := \exp\{\alpha R_\phi(\boldsymbol{x})\}/Z_\phi$:

$$\phi^* = \arg\max_\phi \mathbb{E}_{\boldsymbol{x} \sim p_d}\left[\log q_\phi(\boldsymbol{x})\right]. \tag{7}$$

Given the direct correspondence between the policy $q_{\phi^*}$ in MaxEnt IRL and the policy optimization solution $q_\pi^*$ of Eq.(6), plus the connection between the regularized distribution $q^*$ of PR (Eq.3) and $q_\pi^*$ as built in sec 3.2.1, we can readily link $q^*$ and $q_{\phi^*}$. This motivates to plug $q^*$ in the above maximum likelihood objective to learn the constraint $f_\phi(\boldsymbol{x})$ which is parallel to the reward function $R_\phi(\boldsymbol{x})$. We present the resulting full algorithm in the next section. Table 1 summarizes the correspondence between PR, entropy regularized policy gradient, and maximum entropy IRL.

## 4 Algorithm

We have formally related PR to the RL methods. With the unified view of these approaches, we derive a practical algorithm for arbitrary learnable constraints on any deep generative models. The algorithm alternates the optimization of the constraint $f_\phi$ and the generative model $p_\theta$.

### 4.1 Learning the Constraint $f_\phi$

As motivated in section 3.2, instead of directly optimizing $f_\phi$ in the original PR objectives (Eq.5) which can be problematic, we treat $f_\phi$ as the reward function to be induced with the MaxEnt IRL framework. That is, we maximize the data likelihood of $q(\boldsymbol{x})$ (Eq.3) w.r.t $\phi$, yielding the gradient:

$$\begin{aligned} \nabla_\phi \mathbb{E}_{\boldsymbol{x} \sim p_d}\left[\log q(\boldsymbol{x})\right] &= \nabla_\phi\left[\mathbb{E}_{\boldsymbol{x} \sim p_d}\left[\alpha f_\phi(\boldsymbol{x})\right] - \log Z_\phi\right] \\ &= \mathbb{E}_{\boldsymbol{x} \sim p_d}\left[\alpha \nabla_\phi f_\phi(\boldsymbol{x})\right] - \mathbb{E}_{q(\boldsymbol{x})}\left[\alpha \nabla_\phi f_\phi(\boldsymbol{x})\right]. \end{aligned} \tag{8}$$

The second term involves estimating the expectation w.r.t an energy-based distribution $\mathbb{E}_{q(\boldsymbol{x})}[\cdot]$, which is in general very challenging. However, we can exploit the special structure of $q \propto p_\theta \exp\{\alpha f_\phi\}$ for efficient approximation. Specifically, we use $p_\theta$ as the proposal distribution, and obtain the importance sampling estimate of the second term as following:

$$\begin{aligned} \mathbb{E}_{q(\boldsymbol{x})}\left[\alpha \nabla_\phi f_\phi(\boldsymbol{x})\right] &= \mathbb{E}_{\boldsymbol{x} \sim p_\theta(\boldsymbol{x})}\left[\frac{q(\boldsymbol{x})}{p_\theta(\boldsymbol{x})} \cdot \alpha \nabla_\phi f_\phi(\boldsymbol{x})\right] \\ &= 1/Z_\phi \cdot \mathbb{E}_{\boldsymbol{x} \sim p_\theta(\boldsymbol{x})}\left[\exp\{\alpha f_\phi(\boldsymbol{x})\} \cdot \alpha \nabla_\phi f_\phi(\boldsymbol{x})\right]. \end{aligned} \tag{9}$$

Note that the normalization $Z_\phi = \int p_\theta(\boldsymbol{x}) \exp\{\alpha f_\phi(\boldsymbol{x})\}$ can also be estimated efficiently with MC sampling: $\hat{Z}_\phi = 1/N \sum_{\boldsymbol{x}_i} \exp\{\alpha f_\phi(\boldsymbol{x}_i)\}$, where $\boldsymbol{x}_i \sim p_\theta$. The base generative distribution $p_\theta$ is a natural choice for the proposal as it is in general amenable to efficient sampling, and is close to $q$ as forced by the KL divergence in Eq.(1). Our empirical study shows low variance of the learning process (sec 5). Moreover, using $p_\theta$ as the proposal distribution allows $p_\theta$ to be an implicit generative model (as no likelihood evaluation of $p_\theta$ is needed). Note that the importance sampling estimation is consistent yet biased.

## 4.2 Learning the Generative Model $p_\theta$

Given the current parameter state $(\boldsymbol{\theta} = \boldsymbol{\theta}^t, \boldsymbol{\phi} = \boldsymbol{\phi}^t)$, and $q(\boldsymbol{x})$ evaluated at the parameters, we continue to update the generative model. Recall that optimization of the generative parameter $\boldsymbol{\theta}$ is performed by minimizing the KL divergence in Eq.(4), which we replicate here:

$$\min_\theta \text{KL}(q(\boldsymbol{x})\|p_\theta(\boldsymbol{x})) = \min_\theta -\mathbb{E}_{q(\boldsymbol{x})}\left[\log p_\theta(\boldsymbol{x})\right] + const. \tag{10}$$

The expectation w.r.t $q(\boldsymbol{x})$ can be estimated as above (Eq.9). A drawback of the objective is the requirement of evaluating the generative density $p_\theta(\boldsymbol{x})$, which is incompatible to the emerging *implicit* generative models [40] that only permit simulating samples but not evaluating density.

To address the restriction, when it comes to regularizing implicit models, we propose to instead minimize the *reverse* KL divergence:

$$\min_\theta \text{KL}\left(p_\theta(\boldsymbol{x})\|q(\boldsymbol{x})\right) = \min_\theta \mathbb{E}_{p_\theta}\left[\log \frac{p_\theta \cdot Z_{\phi^t}}{p_{\theta^t} \exp\{\alpha f_{\phi^t}\}}\right]$$
$$= \min_\theta -\mathbb{E}_{p_\theta}\left[\alpha f_{\phi^t}(\boldsymbol{x})\right] + \text{KL}(p_\theta\|p_{\theta^t}) + const. \tag{11}$$

By noting that $\nabla_\theta \text{KL}\left(p_\theta\|p_{\theta^t}\right)|_{\theta=\theta^t} = 0$, we obtain the gradient w.r.t $\boldsymbol{\theta}$:

$$\nabla_\theta \text{KL}\left(p_\theta(\boldsymbol{x})\|q(\boldsymbol{x})\right)|_{\theta=\theta^t} = -\nabla_\theta \mathbb{E}_{p_\theta}\left[\alpha f_{\phi^t}(\boldsymbol{x})\right]|_{\theta=\theta^t}. \tag{12}$$

That is, the gradient of minimizing the reversed KL divergence equals the gradient of maximizing $\mathbb{E}_{p_\theta}\left[\alpha f_{\phi^t}(\boldsymbol{x})\right]$. Intuitively, the objective encourages the generative model $p_\theta$ to generate samples that the constraint function assigns high scores. Though the objective for implicit model deviates the original PR framework, reversing KL for computationality was also used previously such as in the classic wake-sleep method [19]. The resulting algorithm also resembles the adversarial learning in GANs, as we discuss in the next section. Empirical results on implicit models show the effectiveness of the objective.

The resulting algorithm is summarized in Alg.1.

---

**Algorithm 1** Joint Learning of Deep Generative Model and Constraints

---

**Input:** The base generative model $p_\theta(\boldsymbol{x})$
       The (set of) constraints $f_\phi(\boldsymbol{x})$
1: Initialize generative parameter $\boldsymbol{\theta}$ and constraint parameter $\boldsymbol{\phi}$
2: **repeat**
3:     Optimize constraints $\boldsymbol{\phi}$ with Eq.(8)
4:     **if** $p_\theta$ is an implicit model **then**
5:         Optimize model $\boldsymbol{\theta}$ with Eq.(12) along with minimizing original model objective $\mathcal{L}(\boldsymbol{\theta})$
6:     **else**
7:         Optimize model $\boldsymbol{\theta}$ with Eq.(10) along with minimizing $\mathcal{L}(\boldsymbol{\theta})$
8:     **end if**
9: **until** convergence
**Output:** Jointly learned generative model $p_{\theta^*}(\boldsymbol{x})$ and constraints $f_{\phi^*}(\boldsymbol{x})$

---

**Connections to adversarial learning** For implicit generative models, the two objectives w.r.t $\boldsymbol{\phi}$ and $\boldsymbol{\theta}$ (Eq.8 and Eq.12) are conceptually similar to the adversarial learning in GANs [15] and the variants such as energy-based GANs [26, 55, 54, 50]. Specifically, the constraint $f_\phi(\boldsymbol{x})$ can be seen as being optimized to assign lower energy (with the energy-based distribution $q(\boldsymbol{x})$) to real examples from $p_d(\boldsymbol{x})$, and higher energy to fake samples from $q(\boldsymbol{x})$ which is the regularized model of the generator $p_\theta(\boldsymbol{x})$. In contrast, the generator $p_\theta(\boldsymbol{x})$ is optimized to generate samples that confuse $f_\phi$ and obtain lower energy. Such adversarial relation links the PR constraint $f_\phi(\boldsymbol{x})$ to the discriminator in GANs (Table 1). Note that here fake samples are generated from $q(\boldsymbol{x})$ and $p_\theta(\boldsymbol{x})$ in the two learning phases, respectively, which differs from previous adversarial methods for energy-based model estimation that simulate only from a generator. Besides, distinct from the discriminator-centric view of the previous work [26, 54, 50], we primarily aim at improving the generative model by incorporating learned constraints. Last but not the least, as discussed in sec 3.1, the proposed framework and algorithm are more generally and efficiently applicable to not only implicit generative models as in GANs, but also (non-)reparameterizable explicit generative models.

# 5    Experiments

We demonstrate the applications and effectiveness of the algorithm in two tasks related to image and text generation [24], respectively.

| | Method | SSIM | Human |
|---|---|---|---|
| 1 | Ma et al. [38] | 0.614 | — |
| 2 | Pumarola et al. [44] | 0.747 | — |
| 3 | Ma et al. [37] | 0.762 | — |
| 4 | Base model | 0.676 | 0.03 |
| 5 | With fixed constraint | 0.679 | 0.12 |
| 6 | With learned constraint | **0.727** | **0.77** |

**Table 2:** Results of image generation on Structural Similarity (SSIM) [52] between generated and true images, and human survey where the full model yields better generations than the base models (Rows 5-6) on 77% test cases. See the text for more results and discussion.

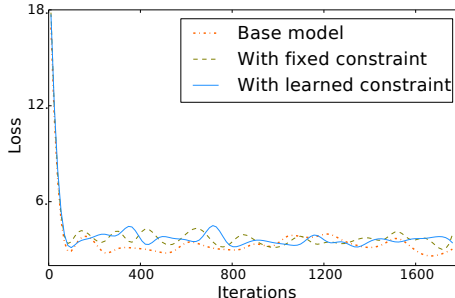

**Figure 2:** Training losses of the three models. The model with learned constraint converges smoothly as base models.

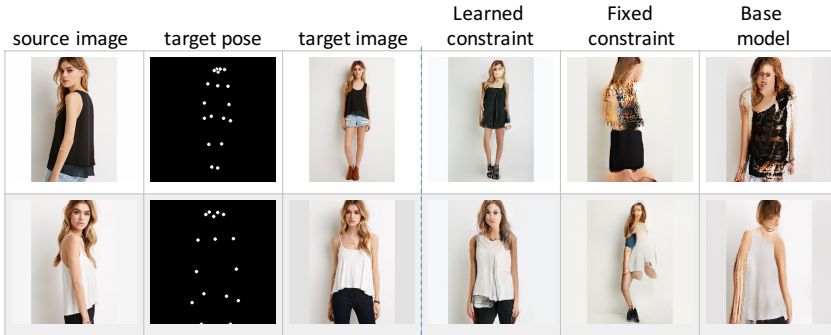

**Figure 3:** Samples generated by the models in Table 2. The model with learned human part constraint generates correct poses and preserves human body structure much better.

## 5.1    Pose Conditional Person Image Generation

Given a person image and a new body pose, the goal is to generate an image of the same person under the new pose (Figure 1, left). The task is challenging due to body self-occlusions and many cloth and shape ambiguities. Complete end-to-end generative networks have previously failed [37] and existing work designed specialized generative processes or network architectures [37, 44, 38]. We show that with an added body part consistency constraint, a plain end-to-end generative model can also be trained to produce highly competitive results, significantly improving over base models that do not incorporate the problem structure.

**Setup.** We follow the previous work [37] and obtain from DeepFashion [35] a set of triples (source image, pose keypoints, target image) as supervision data. The base generative model $p_\phi$ is an implicit model that transforms the input source and pose directly to the pixels of generated image (and hence defines a Dirac-delta distribution). We use the residual block architecture [51] widely-used in image generation for the generative model. The base model is trained to minimize the L1 distance loss between the real and generated pixel values, as well as to confuse a binary discriminator that distinguishes between the generation and the true target image.

**Knowledge constraint.** Neither the pixel-wise distance nor the binary discriminator loss encode any task structures. We introduce a structured consistency constraint $f_\phi$ that encourages each of the body parts (e.g., head, legs) of the generated image to match the respective part of the true image. Specifically, the constraint $f_\phi$ includes a human parsing module that classifies each pixel of a person image into possible body parts. The constraint then evaluates cross entropies of the per-pixel part

| | Model | Perplexity | Human |
|---|---|---|---|
| 1 | Base model | 30.30 | 0.19 |
| 2 | With binary D | 30.01 | 0.20 |
| 3 | With constraint updated in M-step (Eq.5) | 31.27 | 0.15 |
| 4 | With learned constraint | **28.69** | **0.24** |

**Table 3:** Sentence generation results on test set perplexity and human survey. Samples by the full model are considered as of higher quality in 24% cases.

| | | |
|---|---|---|
| \_\_\_\_\_ | acting | \_\_\_\_\_ |
| the | acting | is the acting . |
| the | acting | is also very good . |
| | | out of 10 . |
| | 10 | out of 10 . |
| I will give the movie 7 | | out of 10 . |

**Table 4:** Two test examples, including the template, the sample by the base model, and the sample by the constrained model.

distributions between the generated and true images. The average negative cross entropy serves as the constraint score. The parsing module is parameterized as a neural network with parameters $\phi$, pre-trained on an external parsing dataset [14], and subsequently adapted within our algorithm jointly with the generative model.

**Results.** Table 2 compares the full model (with the learned constraint, Row 6) with the base model (Row 4) and the one regularized with the constraint that is fixed after pre-training (Row 5). Human survey is performed by asking annotators to rank the quality of images generated by the three models on each of 200 test cases, and the percentages of ranked as the best are reported (Tied ranking is treated as negative result). We can see great improvement by the proposed algorithm. The model with fixed constraint fails, partially because pre-training on external data does not necessarily fit to the current problem domain. This highlights the necessity of the constraint learning. Figure 3 shows examples further validating the effectiveness of the algorithm.

In sec 4, we have discussed the close connection between the proposed algorithm and (energy-based) GANs. The conventional discriminator in GANs can be seen as a special type of constraint. With this connection and given that the generator in the task is an implicit generative model, here we can also apply and learn the structured consistency constraint using GANs, which is equivalent to replacing $q(\boldsymbol{x})$ in Eq.(8) with $p_\theta(\boldsymbol{x})$. Such a variant produces a SSIM score of 0.716, slightly inferior to the result of the full algorithm (Row 6). We suspect this is because fake samples by $q$ (instead of $p$) can help with better constraint learning. It would be interesting to explore this in more applications.

To give a sense of the state of the task, Table 2 also lists the performance of previous work. It is worth noting that these results are not directly comparable, as discussed in [44], due to different settings (e.g., the test splits) between each of them. We follow [37, 38] mostly, while our generative model is much simpler than these work with specialized, multi-stage architectures. The proposed algorithm learns constraints with moderate approximations. Figure 2 validates that the training is stable and converges smoothly as the base models.

## 5.2 Template Guided Sentence Generation

The task is to generate a text sentence $\boldsymbol{x}$ that follows a given template $\boldsymbol{t}$ (Figure 1, right). Each missing part in the template can contain arbitrary number of words. This differs from previous sentence completion tasks [9, 57] which designate each masked position to have a single word. Thus directly applying these approaches to the task can be problematic.

**Setup.** We use an attentional sequence-to-sequence (seq2seq) [3] model $p_\theta(\boldsymbol{x}|\boldsymbol{t})$ as the base generative model for the task. Paired (template, sentence) data is obtained by randomly masking out different parts of sentences from the IMDB corpus [8]. The base model is trained in an end-to-end supervised manner, which allows it to memorize the words in the input template and repeat them almost precisely in the generation. However, the main challenge is to generate meaningful and coherent content to fill in the missing parts.

**Knowledge constraint.** To tackle the issue, we add a constraint that enforces matching between the generated sentence and the ground-truth text in the missing parts. Specifically, let $\boldsymbol{t}_-$ be the masked-out true text. That is, plugging $\boldsymbol{t}_-$ into the template $\boldsymbol{t}$ recovers the true complete sentence. The constraint is defined as $f_\phi(\boldsymbol{x}, \boldsymbol{t}_-)$ which returns a high score if the sentence $\boldsymbol{x}$ matches $\boldsymbol{t}_-$ well. The actual implementation of the matching strategy can vary. Here we simply specify $f_\phi$ as another seq2seq network that takes as input a sentence $\boldsymbol{x}$ and evaluates the likelihood of recovering $\boldsymbol{t}_-$ —This

is all we have to specify, while the unknown parameters $\phi$ are learned jointly with the generative model. Despite the simplicity, the empirical results show the usefulness of the constraint.

**Results.** Table 3 shows the results. Row 2 is the base model with an additional binary discriminator that adversarial distinguishes between the generated sentence and the ground truth (i.e., a GAN model). Row 3 is the base model with the constraint learned in the direct way through Eq.(5). We see that the improper learning method for the constraint harms the model performance, partially because of the relatively low-quality model samples the constraint is trained to fit. In contrast, the proposed algorithm effectively improves the model results. Its superiority over the binary discriminator (Row 2) shows the usefulness of incorporating problem structures. Table 4 demonstrates samples by the base and constrained models. Without the explicit constraint forcing in-filling content matching, the base model tends to generate less meaningful content (e.g., duplications, short and general expressions).

# 6 Discussions: Combining Structured Knowledge with Black-box NNs

We revealed the connections between posterior regularization and reinforcement learning, which motivates to learn the knowledge constraints in PR as reward learning in RL. The resulting algorithm is generally applicable to any deep generative models, and flexible to learn the constraints and model jointly. Experiments on image and text generation showed the effectiveness of the algorithm.

The proposed algorithm, along with the previous work (e.g., [21, 22, 18, 36, 23]), represents a general means of adding (structured) knowledge to black-box neural networks by devising *knowledge-inspired losses/constraints* that drive the model to learn the desired structures. This differs from the other popular way that embeds domain knowledge into *specifically-designed neural architectures* (e.g., the knowledge of translation-invariance in image classification is hard-coded in the conv-pooling architecture of ConvNet). While the specialized neural architectures can usually be very effective to capture the designated knowledge, incorporating knowledge via specialized losses enjoys the advantage of generality and flexibility:

- **Model-agnostic**. The learning framework is applicable to neural models with any architectures, e.g., ConvNets, RNNs, and other specialized ones [21].

- **Richer supervisions**. Compared to the conventional end-to-end maximum likelihood learning that usually requires fully-annotated or paired data, the knowledge-aware losses provide additional supervisions based on, e.g., structured rules [21], other models [18, 22, 53, 20], and datasets for other related tasks (e.g., the human image generation method in Figure 1, and [23]). In particular, [23] leverages datasets of sentence sentiment and phrase tense to learn to control the both attributes (sentiment and tense) when generating sentences.

- **Modularized design and learning**. With the rich sources of supervisions, design and learning of the model can still be simple and efficient, because each of the supervision sources can be formulated independently to each other and each forms a separate loss term. For example, [23] *separately* learns two classifiers, one for sentiment and the other for tense, on two *separate* datasets, respectively. The two classifiers carry respective semantic knowledge, and are then *jointly* applied to a text generation model for attribute control. In comparison, mixing and hard-coding multiple knowledge in a single neural architecture can be difficult and quickly becoming impossible when the number of knowledge increases.

- **Generation with discrimination knowledge**. In generation tasks, it can sometimes be difficult to incorporate knowledge directly in the generative process (or model architecture), i.e., defining *how to generate*. In contrast, it is often easier to instead specify a evaluation metric that measures the quality of a given sample in terms of the knowledge, i.e., defining *what desired generation is*. For example, in the human image generation task (Figure 1), evaluating the structured human part consistency could be easier than designing a generator architecture that hard-codes the structured generation process for the human parts.

It is worth noting that the two paradigms are not mutually exclusive. A model with knowledge-inspired specialized architecture can still be learned by optimizing knowledge-inspired losses. Different types of knowledge can be best fit for either architecture hard-coding or loss optimization. It would be interesting to explore the combination of both in the above tasks and others.

**Acknowledgment** This material is based upon work supported by the National Science Foundation grant IIS1563887. Any opinions, findings and conclusions or recommendations expressed in this material are those of the author(s) and do not necessarily reflect the views of the National Science Foundation.

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
