[Reviews · NeurIPS 2018]

Reviewer 1



Summary: The paper proposes a way to incorporate constraints into the learning of generative models through posterior regularization. In doing so, the paper draws connections between posterior regularization and policy optimization. One of the key contributions of this paper is that the constraints are modeled as extrinsic rewards and learned through inverse reinforcement learning. The paper studies an interesting and very practical problem and the contributions are substantial. The writing could definitely be made clearer for Sections 3 and 4, where the overloaded notation is often hard to follow. I have the following questions: 1. Since generative models can be easily sampled from, why not introduce an additional term E_p[f(x)] and minimize L(theta) - E_p[f(x)]. The only explanation I find for introducing q in the paper is “for efficient optimization”. I don’t see theoretically or empirically any reason why the optimization will suffer without q. When f is learned as well, this reduces to energy-based GAN setting discussed in 210-221. I would be curious to see theoretical and/or empirical evidence in support of the approach proposed in the paper. 2. L169-171 needs more clarification. For a self contained paper, the link between q_phi* and q_pi* seems like a key detail that’s missing from the paper. In lines 149-156, the paper establishes a link between q* and q_pi*. In Eq. (8) however, they substitute q_phi with the expression for q*. What’s the justification? 3. Bayesian optimization over the latent space of a generative model is another approach to generate objects with desired properties. It has successfully been applied for generating molecules [1, 2]. Would be good to include a discussion on the merits of either approaches. With regards to the experiments, it’s hard to comment on the significance of the results since the paper claims that the baselines from prior work are not directly comparable (which I agree). I believe energy-based GANs (as mentioned in pt.1 above) and potentially Bayesian optimization techniques (if applicable) would have been good baselines to include. [1] R. Gómez-Bombarelli, D. Duvenaud, J. M. Hernández-Lobato, J. Aguilera-Iparraguirre, T. D. Hirzel, R. P. Adams, and A. Aspuru-Guzik. Automatic chemical design using a data-driven continuous representation of molecules. arXiv preprint arXiv:1610.02415, 2016. [2] M. J. Kusner, B. Paige, and J. M. Hernández-Lobato. Grammar variational autoencoder. arXiv preprint arXiv:1703.01925, 2017. --- --- Thanks for the response. I would highly encourage the authors to include results for the energy based models in the final version, since I believe that to be the closest baseline to this work.

Reviewer 2



The paper proposes a neat approach on including knowledge constraints in deep generative models, namely, GANs. The idea relies on applying Reinforcement Learning (RL) analogy for learning knowledge constraints that has high potential in many applications, such as, image analysis, medical imaging, text analysis etc. The authors explain in detail how the widely used Posterior Regularization (PR) framework for non-learnable knowledge constraints relates to RL. Further, they present how RL could be applied to learn knowledge constraints. The problem is non-trivial and learning the whole approach end-to-end is challenging. However, the authors propose a bunch of approximations that make it happen, e.g.: - approximation of the normalizing constant; - learning p_{\theta} using the reversed KL similarly to the wake-sleep algorithm. I find all these techniques properly chosen and reasonable. The experiments are clear and well performed and show usefulness of the proposed approach. Both image analysis and text analysis are convincing applications. In the future, the method could be used in more challenging problems like medical imaging where having knowledge constraints is very natural. Remarks: 1) The paper lacks a detailed explanation of training process and architectures used in the experiment. I highly recommend including it in the appendix in order to allow the community to reproduce the results. 2) The authors use Eq. (9) to learn the constraints, namely, MC samples from the distribution p_{\theta}. I fully agree that this is a "natural choice", as stated by the authors (line 189), but there are other techniques that could be utilized like REBAR or RELAX. Did the authors consider other approaches? 3) Lines 82-84: The sentences were used in earlier sections.

Reviewer 3



The paper introduces connection between posterior regularization and entropy/KL regularized RL. Through this connection, it proposes to adapt the posterior constraint functions on real data, and shows that the with learned constraints it leads to significantly better generative models compared to the baseline or fixed constraint. It also provides a variant for implicit generative models, where the KL is reversed when fitting the generative model. The results are promising, and the method seems to be applicable for different generative datasets. A question is if adapting constraints on real data could lose its original benefits of constraining the data through high-level feature matching. If the neural networks are very powerful, even if they are initialized with pre-trained parsers etc, could they unlearn them and learn arbitrary statistics that aren’t semantically meaningful? Through constraining specific network architectures and regularization with respect to other tasks, this may be avoided, but it would be good to include some discussions. For example, can you just initialize a specific network architecture with some random weights, and see if it will learn good constraints purely from observed data (without pre-training on external dataset/task)? Comments: -in section 4.1, another benefit of using p_\theta as proposal is that it doesn’t require likelihood evaluation, which is not tractable for implicit models. It’s also good to comment it’s a consistent but biased estimator. -Given it’s not sequential decision making problem, such close connection to RL isn’t very necessary. You may just discuss it from the perspectives of KL minimization/calculus of variations.